# High Resolution and Automatable Cytogenetic Biodosimetry Using In Situ Telomere and Centromere Hybridization for the Accurate Detection of DNA Damage: An Overview

**DOI:** 10.3390/ijms24065699

**Published:** 2023-03-16

**Authors:** Radhia M’Kacher, Bruno Colicchio, Steffen Junker, Elie El Maalouf, Leonhard Heidingsfelder, Andreas Plesch, Alain Dieterlen, Eric Jeandidier, Patrice Carde, Philippe Voisin

**Affiliations:** 1Cell Environment DNA Damage R&D, Genopole, 91000 Evry-Courcouronnes, France; 2IRIMAS, Institut de Recherche en Informatique, Mathématiques, Automatique et Signal, Université de Haute-Alsace, 69093 Mulhouse, France; 3Institute of Biomedicine, University of Aarhus, DK-8000 Aarhus, Denmark; 4MetaSystems GmbH, Robert-Bosch-Str. 6, D-68804 Altlussheim, Germany; 5Laboratoire de Génétique, Groupe Hospitalier de la Région de Mulhouse Sud-Alsace, 69093 Mulhouse, France; 6Department of Hematology, Institut Gustave Roussy, 94804 Villejuif, France

**Keywords:** cytogenetic biodosimetry, radioprotection, telomere, centromere, chromosomal instability

## Abstract

In the event of a radiological or nuclear accident, or when physical dosimetry is not available, the scoring of radiation-induced chromosomal aberrations in lymphocytes constitutes an essential tool for the estimation of the absorbed dose of the exposed individual and for effective triage. Cytogenetic biodosimetry employs different cytogenetic assays including the scoring of dicentrics, micronuclei, and translocations as well as analyses of induced premature chromosome condensation to define the frequency of chromosome aberrations. However, inherent challenges using these techniques include the considerable time span from sampling to result, the sensitivity and specificity of the various techniques, and the requirement of highly skilled personnel. Thus, techniques that obviate these challenges are needed. The introduction of telomere and centromere (TC) staining have successfully met these challenges and, in addition, greatly improved the efficiency of cytogenetic biodosimetry through the development of automated approaches, thus reducing the need for specialized personnel. Here, we review the role of the various cytogenetic dosimeters and their recent improvements in the management of populations exposed to genotoxic agents such as ionizing radiation. Finally, we discuss the emerging potentials to exploit these techniques in a wider spectrum of medical and biological applications, e.g., in cancer biology to identify prognostic biomarkers for the optimal triage and treatment of patients.

## 1. Introduction

Biological dosimetry refers to the quantitative estimation of an absorbed dose of radiation in individuals exposed to ionizing radiation [1]. Radiation exposure may occur due to the geological environment, medical diagnostics and therapy, occupation in radiation facilities or the nuclear industry, or large-scale incidents such as accidents in nuclear industries, nuclear tests, fallouts, nuclear terrorism, or dirty bombs [2,3,4]. Individuals involved in large-scale incidents often do not have personal dosimeters. In cases of exposures requiring dose reconstruction, regardless of the circumstances, biological dosimetry becomes a valuable tool in the accurate assessment of the absorbed radiation dose in the shortest possible time for successful and effective triage and medical management.

The DNA-damaging effects of genotoxic stress such as ionizing radiation may result in chromosome aberrations that can be used as a biomarker, i.e., a biological endpoint to indicate an earlier event as a result of exposure [5]. Biological dosimetry based on cytogenetic assays, termed cytogenetic biodosimetry, has, for decades, been the most extensively studied system. The assays are usually performed on peripheral blood lymphocytes (PBL) because they are easily accessible. The assays are based on the quantification of radiation-induced chromosomal aberrations such as chromosome dicentrics (DC), cytokinesis-block micronuclei (CBMN), premature chromosome condensation (PCC), fragments/rings, or fluorescence in situ hybridization (FISH) for chromosome painting to monitor, e.g., translocations. Three of the techniques, i.e., DC assay (DCA), CBMN, and FISH have now been standardized for monitoring and quantifying the resulting chromosome aberrations in PBL. One of the assays, the DCA, is considered the “gold standard” for biological dosimetry in radiation emergency medicine by IAEA and other international agencies due to the radio-specificity of DCs. Of note, DCA has proven its utility in past large-scale nuclear accidents or accidents such as the Chernobyl accident in 1986 [6].

The assessment of individual doses of exposure in a population should be carried out within the shortest possible time for effective triage and medical intervention, especially in the event of a large-scale radiation incident involving a sizable population group. However, cytogenetic assays for scoring chromosome aberrations suffer from methodological limitations because most of them require a cell culture step, followed by cytological preparations of the cells, and finally scoring under microscopes. Therefore, the time required for performing such biodosimetric measurements and obtaining a result becomes longer than desirable (i.e., >50 h) [1,7,8,9]. The situation is even more challenging when a large number of samples need to be processed, e.g., in a large-scale nuclear incident. Further efforts are thus required to improve methods for scoring radiation-induced chromosome aberrations.

The introduction of peptide nucleic acid (PNA) probes to the scoring of radiation-induced DNA damage has opened new horizons for cytogenetic biodosimetry. Such probes have the advantage of short hybridization times, high specificity, and signal intensity, in addition to low cost [10]. By employing telomere and centromere-specific PNA probes (TC staining), we have obtained significant advances in the scoring of dicentric and ring chromosomes—the best biomarkers of radiation-induced DNA damage—on metaphases from mitosis-induced lymphocytes [10,11] and also on premature chromosome condensation (PCC) in nonstimulated lymphocytes [12,13]. Furthermore, the application of TC staining to the CBMN assay has vastly improved the identification of micronucleus formation, thus increasing the applicability of this assay [14]. Thus, not only have these improvements enhanced the precision of the cytogenetic assays but also obviated the need for personnel with high expertise [15,16].

More recently, as a result of the improvements in these techniques, their applicability has been explored in other fields of biology and their use by other biological non-DNA ionizing radiation assessors have been proposed. Thus, with the advance of cytogenetic techniques, they are not restricted to radiation biology and are not only applicable for the assessment of the effects of any genotoxic agent but also intrinsic defects in maintaining DNA integrity due to pathological gene variants. For instance, chromosomal instability is a prerequisite of ongoing cellular transformation and cancer development. These significantly improved cytogenetic techniques should constitute a valuable tool for the improved characterization of karyotypes in cancer patients, and hence, focused individual treatment and follow-up [17,18].

Here, we present an overview of the advantages of employing TC staining in current techniques used for cytogenetic biodosimetry, i.e., the shorter processing time, higher resolution, accuracy, and automatable recording, and we discuss the clinical utility of these approaches and their great potential in biological dosimetry.

## 2. Cytogenetic Markers of Irradiation

Cytogenetic biodosimetric assays are based on several biological endpoints essentially related to chromosomal aberrations that are consequences of DNA damage in peripheral blood lymphocytes (Figure 1). The markers of irradiation can be classified into three groups:

(1)Direct molecular consequences such as single and double-strand breaks in the DNA molecule detected by fluorochrome-labeled antibodies specific for, e.g., γH2AX, MR11 proteins, or p53-binding protein (53BP1) that are markers of double-strand breaks (DSB) [19,20]. The preparation and analysis of the immunostainings can be automated [4,21]. The proteins are useful markers of DSB (Table 1) in analyses of putative biological effects of low doses of irradiation [22] and for the triage of exposed populations [23,24]. In addition, immunostaining can be used to assess interindividual sensitivity to radiation [25]. However, the specificity of the abovementioned markers to irradiation, the lack of stability of the fluorescent signals over time, the standardization of the technique, and the interindividual variation constitute the main drawbacks for their use in biological dosimetry [26,27].(2)The kinetics of DNA break repair is monitored in prematurely condensed chromosomes (PCCs) in the form of single-stranded filaments. PCCs can be observed by ordinary light microscopy or by fluorescence microscopy after staining with appropriate DNA dyes [28,29]; however, the techniques are subject to significant constraints and low stability of the acentric breaks over time, which limit their use in particular investigations.(3)The consequences of DNA misrepair on the integrity of chromosomes at the time of the first postirradiation cell division or during the second interphase. Such failures resulting in more or less stable dicentrics, translocations, acentric fragments, or micronuclei can be visualized microscopically by the employment of relevant cytogenetic techniques of high sensitivity, specificity, and accuracy. Moreover, the techniques are automatable and easy to implement.

The long-term consequences of irradiation on DNA integrity have been investigated in populations exposed to irradiation following nuclear accidents or medical treatments, and also in their offspring [30,31,32,33,34,35,36] (Figure 1). Based on the collected data, multiple models of radiation-induced carcinogenesis and the transmission of chromosome aberrations to offspring have been proposed [37].

## 3. TC Staining Allows Insight into Mechanisms of Formation of Chromosome Aberrations

TC staining represents a major advance in not only the sensitivity of detection and classification of complex chromosome aberrations but also in elucidating mechanisms of their formation based on their structural alterations. The staining of telomeres, i.e., the ends of each chromosome, allows a very precise determination of the chromosomal landscape. Centromere staining makes it possible to define the nature of the chromosome aberrations and ultimately predict how sustainable they will be during subsequent cell divisions.

Thus, using TC staining, the accurate nomenclature of chromosomal aberrations can be established: (1) Dicentric chromosome: two centromere signals with four telomere signals; (2) centric ring chromosome: one centromere signal without any telomere signal; (3) acentric ring without telomere and centromere signal; (4) acentric chromosomes with four telomere signals or acentric fusion accompanied by dicentric or ring or translocation formation; (5) acentric chromosome with two telomere signals related to terminal deletion; and (6) acentric chromosome without telomere and centromere signals are related to interstitial deletion (Figure 2).

Moreover, TC staining makes it possible to distinguish between metaphases in the first and second mitotic cell division without any supplementary step related to the cross-hybridization of the centromere signal [11].

Furthermore, the scoring of TC-stained chromosome preparations is fully automatable [11,12,17]. Therefore, we believe that the implementation of TC staining will open new fields of research and diagnosis offering the vastly improved visualization of chromosomes and the identification and definition of their associated aberrations.

## 4. Employment of TC Staining Adds Distinctive Value to Commonly Used Cytogenetic Techniques

Table 1 summarizes the sensitivity of commonly used cytogenetic biological dosimetry techniques and the possible detection of recent or past irradiation as well as the homogeneous and heterogenous exposure (Table 1). 

### 4.1. The “Gold Standard” Technique

Among the various assays employed in cytogenetic biodosimetry, the dicentric chromosome assay (DCA) is considered the “gold standard” in radiation emergency medicine.

It is medically and legally recognized [1,7,8,9] and is approved by the International Atomic Energy Agency (IAEA) because of its proven utility in past large-scale nuclear incidents and accidents [30,38].

A dicentric chromosome is an abnormal chromosome with two centromeres. It is formed through two double-strand breaks (DSB) and followed by the fusion of two chromosome fragments, each with a centromere, resulting in the formation of a dicentric chromosome together with acentric fragments (Figure 2B) [8].

Dicentric chromosomes, together with centric rings, the latter formed by the fusion of sticky chromosome ends (Figure 2B), occur in peripheral lymphocytes of individuals as a consequence of recent exposure to an external source of ionizing radiation. Their frequency is dose-dependent. However, they are unstable aberrations, and hence their frequencies decrease during subsequent cell divisions. The acentric fragments can also provide an aid in dose estimation, e.g., to substantiate the presence of other aberrations or, especially in cases of partial body exposure, to verify information about the incident.

Dicentrics and centric rings are relatively simple to score and enumerate. Moreover, their frequency is dose-dependent. For those reasons their enumeration in peripheral blood lymphocytes, termed the “dicentric chromosome assay” (DCA), constitutes the basis of the “Gold Standard” technique for estimating biological dosimetry.

The abovementioned chromosomal aberrations can be observed after uniform staining (Giemsa or DAPI) of metaphases of lymphocytes from a blood sample after their cultivation for 48 h in the presence of phytohemagglutinin (PHA), a stimulator of mitosis, followed by a 2–4 h colcemid-induced block of metaphase (Figure 3A). 

Several studies have shown that frequencies of chromosome aberrations are similar in in vitro and in vivo irradiated blood lymphocytes, respectively. Hence, a calibration curve of the dose-effect relationship obtained after the in vitro irradiation of blood is used to estimate the dose of exposure from an irradiation event in vivo [39,40].

The operating conditions are described in numerous publications such as the IAEA technical guides [1,7,8,9]. Key points of the technique to validate its quality and reproducibility are defined in several ISO standards [41,42,43].

By incorporating TC staining into the “gold standard” technique, the identification of normal chromosomes as well as their aberrations has been vastly improved. The visualization of telomere sequences permits the identification of chromosome ends, and visualization of the centromere sequences, the nature of a chromosome, i.e., (*i*) dicentric: chromosome with two centromere signals (Figure 2B); (*ii*) centric ring: circular chromosome without telomere signal and with only centromere signal (Figure 2B); (*iii*) acentric: chromosome without centromere signal and with or without telomere signal (Figure 2 and Figure 3) [10,11]. Thus, TC staining allows the accurate and reliable detection of all unstable chromosomal aberrations in a single analysis. Importantly, the analysis can be made operator independent [11,44]. Last, but not least, software has been developed that allows the satisfactory automation of chromosomal aberrations scoring [11,12,45,46,47,48,49]. Thus, the technique has significantly reduced the labor-intensive and time-consuming burden of enumerating chromosome aberrations in the classical “gold standard” assay. Moreover, this improved technique should also be applied to studies on the effects of any external genotoxic agent and intrinsic cellular factors affecting chromosome stability.

### 4.2. Detection of Translocations

Chromosomal anomalies that are the result of exchanges of genetic material between non-homologous chromosomes are termed translocations (Figure 2B) [50,51,52,53,54,55,56]. The frequency of radiation-induced translocations in a cell population depends on the dose of irradiation [57]. Some of the translocations may persist for decades because they remain compatible with cell division, as documented in biodosimetric studies on A-bomb survivors performed decades after exposure, and on patients observed for decades after their treatment with radiotherapy for ankylosing spondylitis [58]. Thus, in contrast to dicentrics, rings, and fragments that are lost progressively in subsequent mitotic divisions, a translocation is a better biomarker for retrospective dose evaluation when there has been a long delay between exposure and blood sampling.

In the past, the identification and analysis of translocations based on banding techniques required highly skilled staff to assess karyotypes. However, the introduction of the FISH technique has revolutionized this analysis by using one or several different chromosome-specific colors to “paint” each pair of chromosomes. Each chromosome-specific DNA probe is labeled with fluorescent molecules allowing each chromosome pair or group to be visualized individually or collectively. Any exchange of non-homologous chromosomal material can therefore be easily identified. The fastest method at present is to ‘paint’ three pairs of chromosomes, each pair being the same or different in color from another pair [54,55]. However, some imprecision in the detection of the nature of aberrations cannot be completely avoided, such as an incomplete translocation or terminal translocation, and the distinction between translocation and a specific configuration of a dicentric chromosome with both centromeres in close proximity [17,18]. However, by combining telomere and centromere staining with chromosome painting, the scoring of chromosomal aberrations becomes considerably simpler and more reliable. A further advantage of combining these techniques is that the nature of more complex aberrations can be established with high accuracy, as demonstrated in Figure 4.

### 4.3. The Cytokinesis-Blocked Micronucleus Assay (CBMN)

The scoring of micronuclei has been proposed as an alternative to the conventional quantification of chromosomal aberrations because the assay is faster and also easier. Micronuclei are biomarkers not only of DNA damage but also of genomic instability [59]. However, the high baseline frequency of micronuclei in healthy populations has limited the sensitivity and application of the CBMN assay for the follow-up of exposed populations [60].

Micronuclei containing acentric chromosome fragments or chromatid fragments arise due to non- or misrepaired DNA double-strand breaks as a result of exposure to clastogenic agents, e.g., ionizing radiation [61,62]. Micronuclei can also contain whole chromosomes that lag behind and do not attach to the mitotic spindle during the segregation process in anaphase. Such micronuclei arise as a result of exposure to aneugenic agents (e.g., intracellular oxidants and polycyclic aromatic hydrocarbons) [62], and they represent the main fraction (>70%) of spontaneously occurring micronuclei [14,63]. Irrespective of the cause of their origin, these chromosomes or chromosome fragments are subsequently enveloped by a nuclear membrane and appear in the cytoplasm as small nuclei, i.e., micronuclei, separated from the main nucleus (Figure 2C). The assay is based on peripheral lymphocytes stimulated to undergo mitosis by in vitro stimulation with phytohemagglutinin (PHA). Micronuclei can subsequently be observed in interphase cells.

Ionizing radiation is a strong clastogenic agent and thus a potent inducer of micronuclei, most of which lack centromeres. The CBMN assay has now been established as a reliable technique in radiobiology for the assessment of the radiation exposure of medical, occupational, or accidentally exposed individuals.

The enhanced resolution of the CBMN assay has been obtained by including TC staining or only centromere staining [14,63,64,65]. The achievements using that approach are significant: (1) The visualization of both the telomere and the centromere sequences enhances the sensitivity and detection of MN. (2) The determination of the nature of the genotoxic exposure: clastogenic effect (i.e., MN–T with only telomere sequences) and aneugenic (i.e., MN–TC with telomere and centromere staining) (*cf.* Figure 2C and Figure 5). In addition, the introduction of TC staining into the CBMN assay may advance our insight into molecular mechanisms giving rise to dicentric chromosomes due to the improved scoring of long anaphase bridges with TC sequences and short anaphase bridges without any staining (Figure 2C).

### 4.4. Premature Chromosome Condensation (PCC) Assay

Occasionally, it is difficult to obtain sufficient numbers of metaphases in lymphocytes grown in a culture with PHA. An alternative useful method to the gold standard technique is the premature chromosome condensation (PCC) assay that facilitates the visualization of interphase chromatin as a condensed form of chromosomes. Usually, chromosomes condense during mitosis, following a strict order that is under strict cellular controls. However, it is possible to artificially uncouple chromosome condensation from the mitotic sequence, which makes it possible to visualize chromosomes in, e.g., resting PBL. Two different approaches are usually employed to induce PCC: (1) The PCC fusion approach (PCC–CHO), and (2) the chemical induction approach. In the fusion approach, Chinese hamster ovary (CHO) cells in the M-phase of their cell cycle are fused with the target interphase cells using a fusogen-like polyethylene glycol or inactivated Sendai virus. Prior to fusion, mitotic CHO cells have been synchronized and stocks are frozen and stored until used [28,66].

Alternatively, PCC can be induced chemically in PBL at any stage of the cell cycle by treatment with okadaic acid or calyculin, which are specific inhibitors of serine/threonine protein phosphatase [13,67,68]. The technique is very easy to practice, simply by substituting colcemid with either of the two chemicals. Incubation times of only 30 min, sometimes only 5–10 min (in contrast to the 2–4 h colcemid block) induce a sufficient number of chromosomes. The PCC index is usually much higher (>10%) than the mitotic index (1–2% at best), which makes chromosome analysis much easier. In addition, the drug-induced PCC has outstanding merit that allows the metaphase chromatin, i.e., G_1_-, S-, and G_2_-phase chromatin to be visualized as condensed forms of chromosome structure.

The induction of PCC allows the observation of chromosomes with a light microscope not only during mitosis but also in interphase and hence the analysis of (*i*) chromosome breakage and repair after exposure to ionizing radiation or chemical mutagens; (*ii*) DNA duplication; (*iii*) conformational changes during the cell cycle. Thus, PCCs constitute a useful alternative in cases where metaphase analysis may fail [69], e.g., (1) Estimation of a heterogeneous dose when the collected cells are a mixture of unirradiated and highly irradiated cells, delayed by repair mechanisms resulting in a delayed entry into the cell cycle; (2) accidental exposure to high levels of ionizing radiation, which produces a mitotic delay of unknown duration accompanied by significant apoptotic death; or (3) a very low chronic mitotic index, as in some pathological situations or elderly people.

By combining TC staining with the PCC technique for the analysis of DNA damage, the advantages include: (1) The possibility, for the first time, to visualize and score dicentric chromosomes; (2) determining the nature of acentric breaks; and (3) higher precision to monitor the kinetics of formation of chromosomal aberrations (Figure 6). The combination of these two techniques opens new horizons for the PCC technique, not only for the rapid scoring of DNA damage but also in the application of biological dosimetry into radiation emergency medicine and the automation of the analysis [12,68,70].

## 5. Automation of Biological Dosimetry Methods

Following considerable development of novel reagents and the standardization of cytogenetic preparations, the quality of cytogenetic slides has increased progressively to levels that make it feasible to automate the search for metaphases and capture slides. The first attempts to automate the scoring of radiation-induced DNA damage began in the 1990s with the automation of the scoring of micronuclei after uniform DNA staining (Giemsa or DAPI) [71]. The first commercially available system was Metafer MNScore from MetaSystems [72].

Spectacular advances have been achieved in recent years in terms of sensitivity and interface after uniform staining using cytogenetic slides or flow cytometers (DAPI) [21,73,74,75]. However, this automation concerns only the scoring of micronuclei and does not make it possible to differentiate the origin of micronuclei because centromere staining has not yet been implemented. The automation of other DNA damage markers such as anaphase bridges or nuclear buds (NBUDs) is still not relevant despite their crucial importance in biological dosimetry or chromosomal instability fields [76].

The second attempt of automation concerns the scoring of dicentric chromosomes after uniform staining. Reliability had long been hampered by the difficulty of recognizing metaphases by the use of many processors with little software. The first commercially available system was Metafer DCScore from MetaSystems [77,78,79]. A spectacular improvement was achieved in terms of sensitivity, which is essentially due to the introduction of TC staining [11,80,81] and the marketing of more advanced software.

The automation of PCC–CHO following uniform staining [66] and subsequent TC staining [12] has now been initiated; however, the implementation of this technique and its current use requires further technical development.

The automation of the scoring of translocation has not been undertaken so far. However, ongoing work using machine learning approaches after TC staining followed by M-FISH should make it possible to achieve high efficiency in the automation process.

## 6. Application of Cytogenetic Tools in a Wider Spectrum of Fields in Medicine and Biology

Cytogenetic biodosimetry is not limited to ionizing radiation and the follow-up of exposed populations. It is, in fact, applicable in the assessment of the effect of any genotoxic agent as well as that of inherent spontaneous genome instability in, e.g., cancer patients. The need to improve cytogenetic biodosimetry assays has been driven by the need to transfer technology to all cytogenetic laboratories to achieve high sample throughput for the processing of large cohorts of exposed populations as well as in medical settings [17,18,82].

The multiplex (M-FISH) technique is the latest evolution of FISH, which stains sex chromosomes and each pair of autosomes with different colors (1986). However, M-FISH has not been used intensively due to the complexity of the analysis. Recently, some studies started to use this technique [83,84,85,86,87,88]; however, the resolution of chromosomes to identify complex rearrangements can be vastly improved when TC staining with M-FISH [17,18,89].

TC staining followed by multiplex FISH (M-FISH), termed (TC+M-FISH), allows the reliable analysis of both unstable and stable chromosomal aberrations in a single-step analysis (Figure 7). In addition, this approach makes it possible to obtain an accurate karyotype in cases of genome heterogeneity and clonal escape.

The employment of the TC+M-FISH approach in the analysis of the transmission of unstable chromosomal aberrations through multiple cell divisions has permitted not only the reevaluation of the transmission of dicentric chromosomes but also, for the first time, the documentation of the transmission of an acentric chromosome and the involvement of centromere breakpoints in this transmission [44]. Moreover, we have been able to demonstrate the persistence of specific configurations of dicentric chromosomes with both centromeres in close proximity during cell division [44]. Such dicentric chromosome configurations can easily be mistaken for translocation when only using the M-FISH technique because centromeric regions are not visible in the M-FISH assay.

Recently, we demonstrated the clinical utility of the TC+M-FISH technique for the detection of chromosomal aberrations in cancer patients in general, and of dicentric chromosomes in particular [17]. Today, the latter is considered the best cytological biomarker of chromosomal instability in cancer patients [90].

The TC+M-FISH approach permits the establishment of reliable and accurate karyotypes for cancer diagnosis [17,18]. Thus, the application of TC+M-FISH has revealed a much higher frequency of dicentric chromosomes with a specific configuration (two centromeres in close proximity) than has been observed previously by using conventional and molecular cytogenetics.

Moreover, the TC+M-FISH technique makes it possible to detect both stable and unstable aberrations, allowing accurate identification with high sensitivity of all chromosomal aberrations in addition to the putative progress of clonal escape and chromosomal instability. This approach will improve our knowledge not only of the biological effects of chemotherapy or low doses of irradiation but also of the underlying mechanisms of the formation of chromosomal aberrations. It could also be applied, with advantage, in the follow-up of exposed populations with a high risk of developing secondary cancer or late complications after exposure to genotoxic agents.

## 7. Advantages and Limitations of Current Cytogenetic Biomarkers of Ionizing Radiation

Most of the cytogenetic markers that have been used routinely for the past 50 years are primarily indicators of structural or functional damage to cellular components. However, despite their well-documented advantages, several challenges remain to be addressed and solved.

### 7.1. Specificity of Cytogenetic Biodosimetry Markers

It should be borne in mind that none of the chromosomal biomarkers described above are specific for monitoring exposure to ionizing radiation. In fact, they may also reflect the combinatory influence of other stressors such as the prior health status of the individual and genotoxic contamination by pollutants or chemicals that may create severe and long-lasting pathophysiological damage. Consequently, these markers constitute valuable indicators of the stress status of an individual (e.g., DNA repair, chromosome instability, and carcinogenesis). At low and medium doses of ionizing radiation, they may also reflect the subtle association between different additive effects of ionizing radiation such as radiation sensitivity, adaptation, and bystander effect, the consequences of which are not fully understood.

### 7.2. Challenges in Dose Estimation

The spontaneous rate of chromosomal aberrations in the general population is an important factor in cytogenetic biodosimetry because it can influence the interpretation of data. Thus, DNA damage can also be induced by occupational activities, lifestyle, and environmental factors [91]. Furthermore, there are variations in natural, terrestrial, and cosmic radioactivity, which differ from region to region and from country to country [92]. Finally, the worldwide increase in exposure to magnetic fields such as the use of mobile phones and the multiplicity of the employment of ionizing radiation in industry and medicine may have contributed to significant variations [93]. The latest evaluation of the frequencies of spontaneous chromosome aberrations dates back several decades. Since then, a vast amount of insight has accumulated into the causes and frequencies of chromosome aberrations. Moreover, significant technical improvements in their detection have been achieved. Therefore, the re-examination of the frequency of spontaneous chromosomal aberrations in the general population is urgently needed. For that purpose, an automated TC+M-FISH approach would be most relevant.The distribution of aberrations according to age and sex is not clear and differs from one study to another. Recently, several studies have demonstrated the difference in genotoxic stress response according to sex [94,95,96]. Notably, the used dose-response curves did not take into account the age or sex of the exposed population.The interpretation of complex chromosomal rearrangements in the estimation of the absorbed dose has always been challenging in biological dosimetry. A significant correlation has been found between the formation of complex chromosomal rearrangements and the clinical outcome of patients treated with radiotherapy [97,98]. The presence of these kinds of aberrations has also been correlated with interindividual radiation sensitivity and genomic instability [99]. The lack of analysis of complex chromosomal rearrangement in a large cohort of an exposed population using a sensitive technique did not make it possible to advance our knowledge regarding their formations and their interpretations.The interpretation of “Rogue cells” in the analysis of chromosomal aberrations and the estimation of the dose after exposure is still unclear. “Rogue cells” are cells with multiple and complex chromosomal aberrations (e.g., dicentric, tricentric, translocations, insertions, deletions, and acentric chromosomes) related to the activation of viral infection [100,101]. A significant increase in induced chromosomal aberrations has been detected in the presence of rogue cells [102,103,104,105]. Further studies are needed to investigate the role of viral infection in the formation of radiation-induced chromosomal aberrations.

### 7.3. Relevant Questions for Cytogenetic Biological Dosimetry Assays

Although research over the last 50 years on cytogenetic biomarkers in biodosimetry has sufficiently covered their operational capacities, there still persist some gray areas that need to be addressed and resolved to improve the assessment of potential genotoxic exposures [52,106]:
-How can we validly relate lymphocyte lifetime and (re)circulation to partial exposure and thus introduce useful correction factors in the estimation of an absorbed dose of ionizing radiation?-What value can be attributed to translocation analysis in the concept of dose, especially decades or years after a potential exposure?-How do we coordinate all of the biological and biophysical markers available to the dosimetrist into a coherent entity, integrating the concept of multiparameter analyses?-How do we integrate new developments (genomics, proteomics, and transcriptomics) that, without renovating the current biotechnological landscape, make it possible to associate physiopathology more globally with genomic instability?-How do we integrate modulation phenomena, such as cellular and tissue radiosensitivity, radiation adaptation, and abscopal or bystander effects, that are still incompletely understood?

### 7.4. Internal Exposure

The internal exposure to ionizing radiation, particularly that due to contamination, is classically considered in three specific contexts: (*i*) environmental, to which all people are naturally subject; (*ii*) occupational, concerning certain categories of workers, and finally, (*iii*) iatrogenic, the consequences of which evolve with the level of health care [107]. Biological indicators could be very useful to characterize the occurrence and type of effects observed, as well as the intracellular and intercellular processes activated in response to internal exposure. The first interest is the clinical and epidemiological study of cancers developed by humans after exposure to a radio-contaminant; the second interest is related to clinical treatment and the histological characterization of treated cancers after induction of vectorized radioimmunotherapy.

## 8. New Challenges for the Use of Biological Dosimetry in Detecting Carcinogenesis Susceptibility

The accurate detection of DNA damage is not restricted to cytogenetic biodosimetry after accidental, professional, or medical exposure to ionizing radiation [108]. Personalized medicine also needs this approach for the development of new biomarkers of DNA repair deficiency in order to propose a specific treatment for individual patients. The clinical utility of these techniques—particularly the need for sensitive and specific biomarkers—is justified by the stratification of patients regarding the deleterious consequence of chromosomal instability [108,109]. The introduction of TC staining in the cytogenetic biological dosimetry approach and the automation of the scoring of all chromosomal aberrations must lead to the creation of databases allowing sensitive and specific stratification of exposed populations and/or patients.

The second challenge is related to the impact of low-dose effects on chromosome instability. Improved detection of DNA damage by introducing TC staining makes it possible to reveal genetic consequences for doses below 0.1 Gy. Their application, including telomere analysis to the field of low-dose exposures, could add significant advantages to our understanding of such effects.

The third challenge is paradoxically to have the best-standardized response for various types of irradiation, dose rate, and genetic parameters in any laboratory practicing biological dosimetry assessment. This apparently simple feature, although required for better efficiency, is not always obtainable—and the reasons for that are not always obvious. The introduction of TC staining in the automatic scoring of DNA damage has rendered the analysis independent of the level of expertise of the operator. In addition, high reproducibility of the scoring has been achieved. The systematic use of this technique instead of the classical scoring of chromosome aberrations could greatly improve the respective standardizations.

Finally, the possible fusion between the “gold standard” technique (dicentric chromosome), FISH or M-FISH technique (translocation), and CBMN assay (micronuclei) will represent an important technical challenge that will surely lead to wider applications for sensitive and reliable dose estimation [110,111].

While biodosimetry using traditional cytogenetic assays is useful for dose assessments following accidental exposure to ionizing radiation, the field of biodosimetry has advanced significantly with expansion into the fields of genomics, proteomics, metabolomics, and transcriptomics [112,113,114,115]. However, advances in the assessment of radiation exposure using those approaches have more relevance to clinical applications for patients undergoing radiation therapy. The information derived from radiation responses may advance personalized care and could help alleviate the damage to normal tissue [116].

## 9. Conclusions

For more than 50 years cytogenetic biodosimetry has been routinely and successfully practiced worldwide by a few specialized laboratories for assessing individual exposure to ionizing radiation, and international networks have been established in preparation for a potential nuclear catastrophe. However, all of the classical techniques applied for these purposes suffer from several limitations: they are unnecessarily time-consuming, their sensitivity and specificity are untimely limited, and they require highly skilled personnel that may be in shortage in the event of a major nuclear catastrophe. By including TC staining together with M-FISH, several of these obstacles may be overcome. Moreover, importantly, the automation of these novel techniques adds to the major achievements provided by the inclusion of TC staining in the current cytogenetic assays. We also propose their use in the general assessment of DNA damage and chromosome instability which are largely underestimated in many diseases. Last but not the least, wider potentials exploiting these techniques are emerging for their application in identifying prognostic biomarkers and guides for the better triage and management of medical patients. The standardization of cytogenetic biological dosimetry approaches permits the use of a unique dose-response curve. The automation of these approaches opens new horizons in the construction of large databases in exposed populations, allowing the precise estimation of risk associated with radiation exposure. This is the role of official instances such as the AIEA and OMS as well as an international research program for the harmonization and standardization of the process such as the RENEB program. We also need companies to manufacture the kits and for standardization and development of analysis software with the overall aim of automating all of the processes.

## Figures and Tables

**Figure 1 ijms-24-05699-f001:**
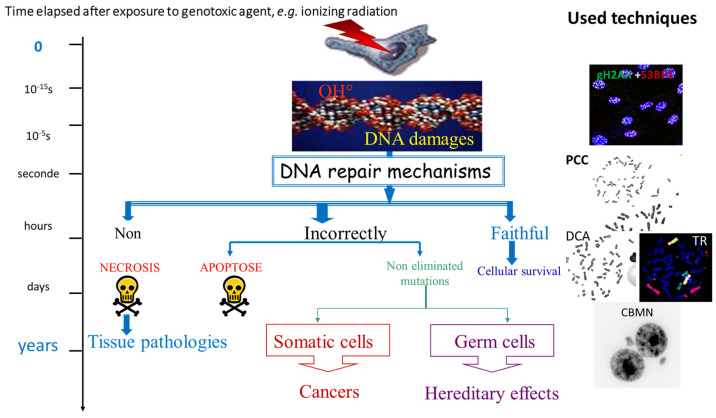
Schematic overview of the consequences of DNA repair mechanisms of varying efficiencies, induced after exposure to genotoxic agents. To the right are depicted examples of cytogenetic preparations using assays employed in cytogenetic biodosimetry. PCC: premature chromosome condensation; DCA: dicentric chromosome assay; TR: scoring of translocations; and MN: micronucleus assay.

**Figure 2 ijms-24-05699-f002:**
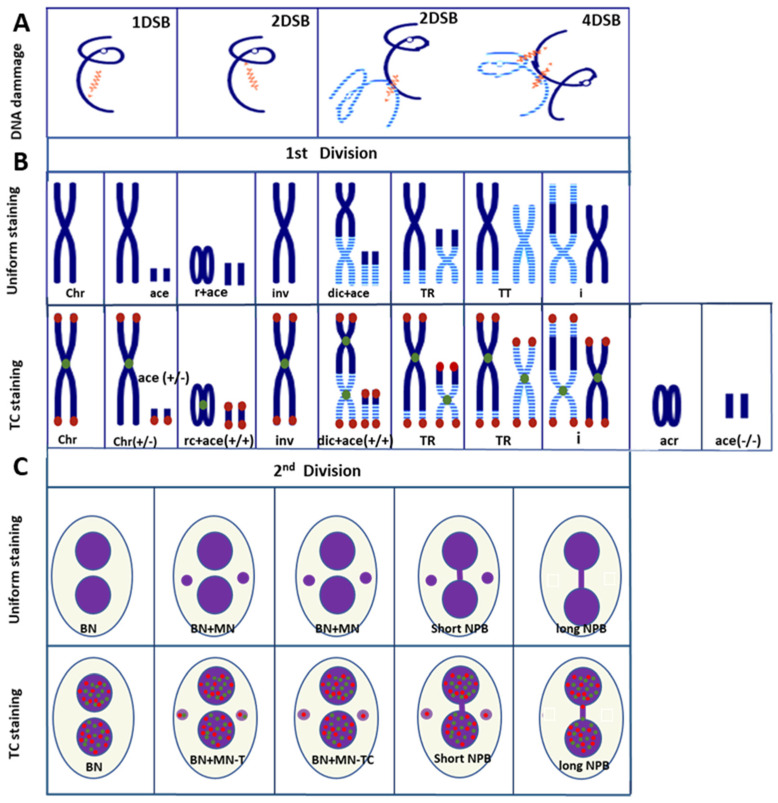
DNA damage and chromosomal aberrations after exposure to ionizing irradiation: (**A**) DNA damage after one, two, or four DSB. (**B**) Chromosomal aberrations in the first mitosis are visualized by uniform staining or TC staining. The latter permits the precise identification of chromosomal aberrations. Ace: acentric fragment; Rc: ring chromosome; Dic: dicentric chromosome; inv: inversion; TR: translocation; I: insertion; Gray: centromere; and red: telomere. (**C**) CBMN assay followed by TC staining for assessing DNA damage permits a distinction between aneugenic and clastogenic exposures. CBMN assay followed by TC staining allows the detection of the mechanisms of the formation of an anaphase bridge: a short anaphase bridge without TC staining related to the presence of a dicentric chromosome with centromere breakpoints and a long anaphase bridge with TC staining related to the presence of dicentric chromosomes with telomere fusions.

**Figure 3 ijms-24-05699-f003:**
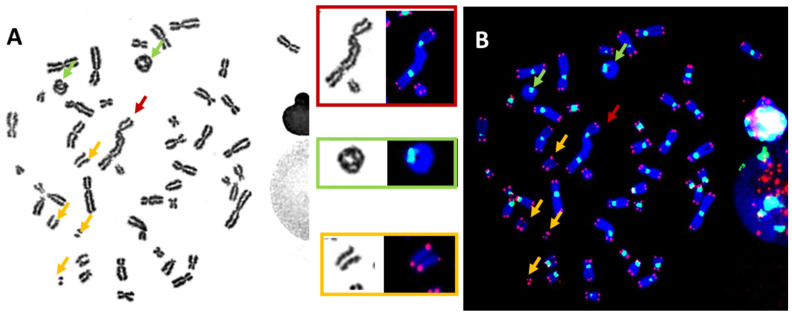
The identification of unstable chromosomal aberrations after the in vitro irradiation of blood lymphocytes (63× magnification) using (**A**) uniform chromosome staining (Giemsa staining) permits the detection of unstable chromosomal aberrations according to their morphologies; (**B**) telomere (red) and centromere (green) staining allow the accurate detection of all unstable chromosome aberrations: (*i*) dicentric chromosome with two green signals and four red signals (top insert), (*ii*) ring chromosomes with one green signal, but no red signals (2nd insert from the top), and (*iii*) acentric fragments with four red signals only (bottom insert).

**Figure 4 ijms-24-05699-f004:**
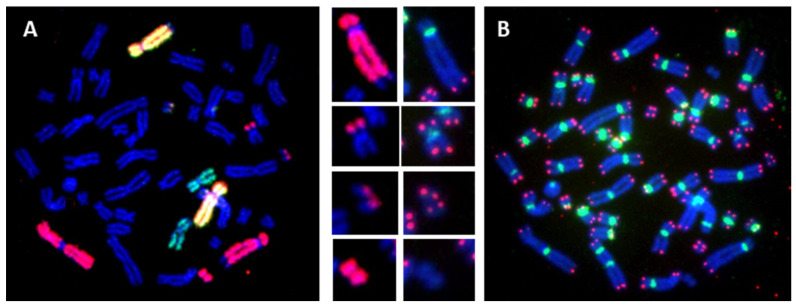
The detection of stable chromosomal aberrations after the in vitro irradiation of blood lymphocytes (63× magnification) using (**A**) chromosome 1 (red), 4 (yellow), and 11 (green) painting and the identification of a complex rearrangement involving chromosome 1 with a reciprocal translocation and of two other undefined aberrations. (**B**) By combining telomere and centromere staining with chromosome painting, the two other aberrations can be defined: telomere deletion (1st insert from the top) was detected in chromosome 1 implicated in this reciprocal translocation (2nd insert from the top); the precise detection of an acentric fragment accompanied the formation of this reciprocal aberration (3rd insert from the top); detection of an acentric ring lacking telomere and centromere staining (bottom insert).

**Figure 5 ijms-24-05699-f005:**
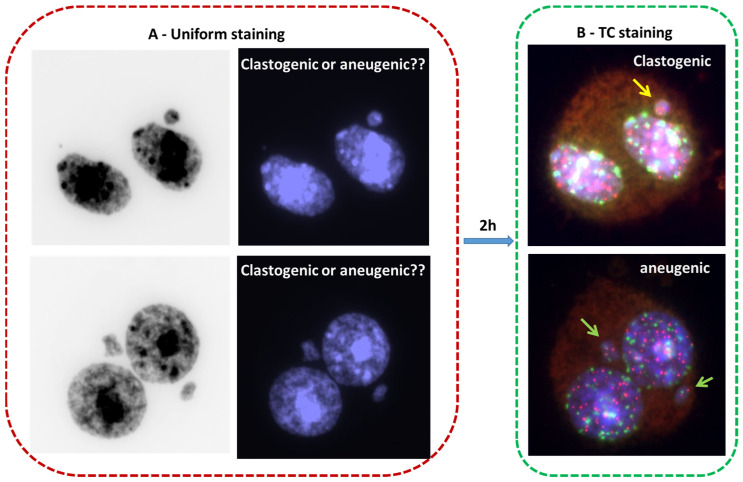
A combination of telomere (red) and centromere (green) staining in the CBMN assay allows the easy identification of the nature of a genotoxic exposure. (**A**) The detection of micronuclei after uniform DNA staining. The staining does not allow the identification of their chromosomal contents. (**B**) Micronuclei with only telomere staining (yellow arrow) as a consequence of exposure to the clastogenic agent, and micronuclei with telomere and centromere staining (green arrow) due to exposure to the aneugenic agent (63× magnification).

**Figure 6 ijms-24-05699-f006:**
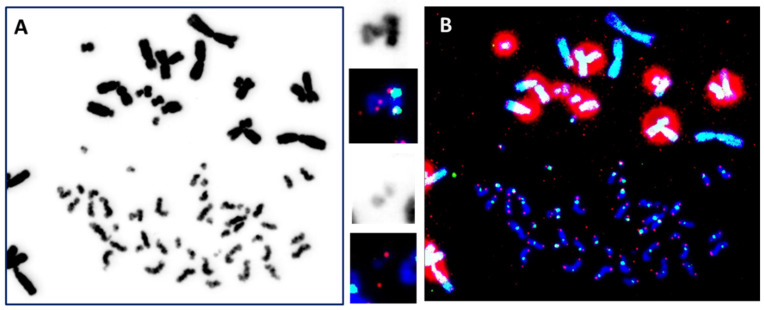
The combination of TC staining with the PCC fusion technique allows the reliable scoring of chromosome fragments and the detection of dicentric chromosomes. (**A**) Conventional DNA staining of PCC permits the scoring of chromosome condensation fragments (63× magnification). (**B**) Simultaneous telomere (red signals) and centromere (green signals) staining permits the detection of dicentric chromosomes (top inserts) and the accurate and consistent detection of acentric chromosomes (bottom inserts). The large red areas represent telomere signals of CHO chromosomes.

**Figure 7 ijms-24-05699-f007:**
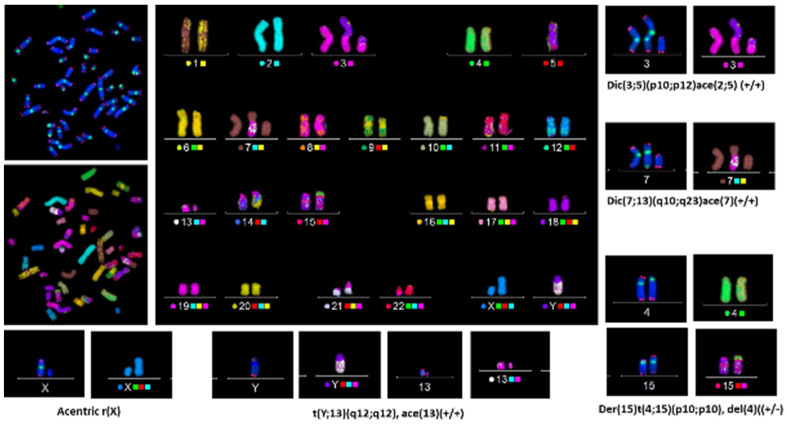
Telomere and centromere staining followed by M-FISH offers the sensitive and accurate detection of all chromosome aberrations, both stable and unstable ones. This approach permits the reliable identification of breakpoints and the assessment of the presence of complex clonal aberrations (63× magnification).

**Table 1 ijms-24-05699-t001:** Different categories of commonly used cytogenetic biological dosimeters/indicators explored for radiation dose reconstruction and possible accidental exposure situations. The question marks indicate that the validity of the test is not yet recognized.

Techniques	Types of Exposure to Ionizing Radiation				Sensitivity
	Recent and Homogeneous Events	Recent and Heterogeneous Events	Past Event	Large-Scale Event	Sensitivity
Dicentric and centric rings	YES	YES	NO	YES	0.1 Gy
Micronuclei	YES	YES	NO	YES	0.3 Gy
Translocations	YES	YES	YES	NO	0.25–0.3 Gy
PCC—CHO	YES	YES	NO	YES	0.1 Gy
PCC—ring	YES	?	NO	?	?

## Data Availability

Not applicable.

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
