# Peer review of "High Resolution and Automatable Cytogenetic Biodosimetry Using In Situ Telomere and Centromere Hybridization for the Accurate Detection of DNA Damage: An Overview"

_ijms, 2023, doi:10.3390/ijms24065699_

Round 1
Reviewer 1 Report
Overall comment
The title of this review entails that the reader will learn how the "recent" introduction of telomere and centromere (TC) painting enhances the accuracy of biodosimetry based on cytogenetic assays that lends themselves to be also automatable. Note that the "recent" is intentionally quoted in brackets because the main motif of this paper is to say that this cytogenetic painting methodology was recent while it is not. However, I regret to say that in this work I found no evidence that automation for scoring the endpoints used in the the classic biodosimetry assays mentioned by the authors (that is chromosome aberrations and micronuclei) was really due to or adjuvanted by TC nor did I find any sensible argument to substantiate the "high-resolution" advertised in the title. This is if by high resolution we intend the use of advanced microscopy techniques pushing the limit of actual detection of DNA damage to the nanometer level, for instance, as is the case with the foci, assay on which I will come back later. Indeed, the authors more than once refer to TC painting as a refined technique, which was the case 15 or more years ago, In general, therefore, I see no novelty or usefulness for this review compared to several ones that have been published and have covered a wider range of more recent developments, which here are just briefly mentioned towards the end. The main shortcoming is that, as actually partially inferred by the quoted references, introduction of TC is dated and whilst indeed allowing for some mechanistical insights in both chromosome aberration and micronuclei genesis, is not related to automation nor to high resolution. And even if it were, nothing really new is here presented.
Punctual comments:
Line 39: On the first page, in the very introduction of the concept of biodosimetry, the authors fail to highlight the essence of this approach, which consists in the attempt to indeed for a "quantitative estimation of the absorbed dose of ionising radiation (IR)", but it is a "retrospective" quantification. This may be obvious to them, but not necessarily to the readers. There occurs here and it also surfaces throughout the review a sort of mixing up contexts, such as listing the sources of possible human exposure to IR or, as seen later, saying that assays used in biodosimetry can also be used in the clinic. This generates confusion because the other topic which is also more or less hinted at is the major issue of estimating the risk for low-dose exposures such as in diagnostics or of secondary cancers in radiotherapy patients. But these are not the topic this review intends to deal with.
Page 2 is a simple explanation of very well-known concepts. Lines 75 and subsequent ones well exemplify my main criticism: the authors present the introduction of TC as something recent and destined to profoundly impact the field of biodosimetry. Except for some self-citations which refer to work that is not so recent anyway (ca. 2014), no other work proving their point is quoted. Also, automation of micronuclei is already implemented by several commercially available softwares that actually work with solid staining: if the issue is improving the rapidity of the CBMN assay, the use of FISH goes in the opposite direction, and this does require more skilled personnel anyway, since micronucleus scoring per se is much less challenging that chromosome aberration classification. And indeed also detection of dicentrics has been object of successful automation using Giemsa or similar-the main reason why the DCA established itself as the gold standars, as reminded by the authors themselves, being the dicentric a generally very easily identifiable structural abnormality.
On page 3, in the paragraph devoted to the cytogenetics markers of radiation, the authors rather bluntly make some statements which are simply not correct regarding the sensitivity and usefulness of the foci assay in biodosimetry. They quote reference 21 to say that the main limitation would be the poor stability of the foci fluorescence signal. However, ref. 21 is wrongly attributed to Lloyd as first author whereas it is Moquet and in that paper the authors, part of the European RENEB collaboration, actually say that the foci assay is a promising tool. Other recent reviews such as those by Jakl et al , Genes, 2020 and Penninckx in NAR Cancer, 2021 do concur in the elevated sensitivity of foci in the low-dose range and for biodosimetry application, thanks for instance to the correlation between their persistence and the severity of induced damage with possible health consequences; other limits exists that are more relevant than the stability of the fluorescence signal, which are shared anyway also by the "classic" assays, and are the lack of a reliable inter-laboratory standardization and a high individual variability.
On page 6, when introducing (with a degree of repetition) the CBMN assay, the authors state than this ensures a higher sensitivity than chromosomal aberrations. If by sensitivity, as I believe is meant in biodosimetry, we mean the lower limit of "measurable"/quantifiable dose through the in vitro constructed dose-response curve, micronuclei do suffer limitations as highlighted already by Bonassi et al in 2001 such as a higher variability, and a marked dependence on confounding factors such as smoking habits, as well as individual age. Going back to the "novelty" of coupling FISH-based TC staining with CBMN assay, Miller et al , Radiat Res, 1992 was already reporting such an approach, which serves a purpose , as the authors correctly say, if one wants to assess whether the micronucleus contains an acentric fragment or a whole chromosome as to explain whether it was mostly due to chemical rather than radiation-induced damage. That is by the way why the micronucleus assay is more general and less specific than chromosome aberrations as an indicator of genotoxicity.
Paragraph 6 on premature chromosome condensation really adds nothing to the current knowledge nor does the following paragraph, where the potential of coupling TC with m-FISH is illustrated. Incidentally, if we want to be rigorous, m-FISH is not the best example for an automatable assay nor for one to be used for rapid biodosimetry triage since it is extremely time consuming, expensive and requires fundamental and highly trained human intervention for the karyotyping part.
Author Response
see attach file

Reviewer 2 Report
This is a nice overview on the use of cytogenetic analysis in the application of biodosimetrynandnresponse to radiation.
The manuscript is in need of the following:
1. A cumulative Table on the proper use of the each methodology for the detection of which specific endpoint, advantages and disadvantages as well as limitations and detection sensitivity for example doses, type of biological samples etc…
2. Mechanistically speaking ,the authors need to discuss how the chromosome breaks and types of chromosome instability and damage result from complex DNA damage like DSBs and non_DSB lesions for example see here : PMCID: PMC5532627
and other references therein.
3. Which are the needs for future advances in the field, how do they see the needed progress and better resolution ?
Author Response
see attach file

Round 2
Reviewer 1 Report
I appreciate the changes made by the authors. The scientific value of the work is not under question here. What I contend is the notion that Telomere and Centromere (TC) hybridization is: a) a high-resolution innovative technique; b) a step forward towards automation; c) an essential tool for accurate biodosimetry. I am well aware of the international consortium revolving around RENEB but the fact remains that TC staining was performed years before it and years before it at least one widely commercially available software was able not only to detect and integrate in a single image signals from a several chromosome probes (including telomere and centromere) but also efficiently count them in an automatic manner. Therefore, i am sorry to say that I continue to see this review as adding nothing new to the field.
Author Response
We thank the reviewer for recognizing the scientific value of our work.
However, if the scientific value of this review is not questioned, I believe that there is a discrepancy between that statement and the reviewer’s additional comments.
MetaSystems were the first to introduce automation of dicentric chromosomes uniformly stained with DAPI or Giemsa. The improved version of DCScore with TC staining was launched on the market shortly before our publication in 2014. I contributed to that work. Indeed, TC staining with DNA probes had been practised many years before, but TC staining was vastly improved when DNA probes were replaced by PNA probes in 2012, because the latter require only very short hybridization times, and yet generate much stronger signals than those using DNA probes. The strong signals and reproducibility of the technique has made it robust and reliable permitting automation of the scoring of chromosomal aberrations.